# Suppression of distracting inputs by visual-spatial cues is driven by anticipatory alpha activity

**Chenguang Zhao**[1,2,3,4☯], **Yuanjun Kong**[1☯], **Dongwei Li**[1], **Jing Huang**[1,2], **Lujiao Kong**[5], **Xiaoli Li**[1,2], **Ole Jensen**[6], **Yan Song** [1]*

**1** State Key Laboratory of Cognitive Neuroscience and Learning &IDG/McGovern Institute for Brain Research, Beijing Normal University, Beijing, China, **2** Center for Cognition and Neuroergonomics, State Key Laboratory of Cognitive Neuroscience and Learning, Beijing Normal University, Zhuhai, China, **3** School of Systems Science, Beijing Normal University, Beijing, China, **4** International Academic Center of Complex Systems, Beijing Normal University, Zhuhai, China, **5** School of Journalism and Communication, Beijing Normal University, Beijing, China, **6** Centre for Human Brain Health, School of Psychology, University of Birmingham, Birmingham, United Kingdom

☯ These authors contributed equally to this work.
* songyan@bnu.edu.cn

**Data Availability Statement:** The materials of these experiments are available at the Open Science Framework (https://osf.io/z9rym/).

**Funding:** This work was supported by grants from the National Natural Science Foundation of China

## Abstract

A growing body of research demonstrates that distracting inputs can be proactively suppressed via spatial cues, nonspatial cues, or experience, which are governed by more than one top-down mechanism of attention. However, how the neural mechanisms underlying spatial distractor cues guide proactive suppression of distracting inputs remains unresolved. Here, we recorded electroencephalography signals from 110 participants in 3 experiments to identify the role of alpha activity in proactive distractor suppression induced by spatial cues and its influence on subsequent distractor inhibition. Behaviorally, we found novel changes in the spatial proximity of the distractor: Cueing distractors far away from the target improves search performance for the target, while cueing distractors close to the target hampers performance. Crucially, we found dynamic characteristics of spatial representation for distractor suppression during anticipation. This result was further verified by alpha power increased relatively contralateral to the cued distractor. At both the between- and within-subjects levels, we found that these activities further predicted the decrement of the subsequent $P_D$ component, which was indicative of reduced distractor interference. Moreover, anticipatory alpha activity and its link with the subsequent $P_D$ component were specific to the high predictive validity of distractor cue. Together, our results reveal the underlying neural mechanisms by which cueing the spatial distractor may contribute to reduced distractor interference. These results also provide evidence supporting the role of alpha activity as gating by proactive suppression.

## Introduction

In daily life, individuals often select a task-relevant target from the surrounding distractors, for which selective attention is required [1]. Competition of simultaneously presented distractors

(No.32271094, No. 31871099 to Y. S.; No. 62201064 to C. Z.; No.32200870 to J. H.), Scientific Technological Innovation 2030—Major Projects (No.2021ZD0204300 to X. L.; No. 2022ZD0211300 to C. Z.).The funders had no role in study design, data collection and analysis, decision to publish, or preparation of the manuscript.

**Competing interests:** The authors have declared that no competing interests exist.

**Abbreviations:** ACC, accuracy; AUC, area under the ROC curve; CTF, channel tuning function; DTD, distance of the target to the distractor location; EM, encoding model; ERP, event-related potential; ES, efficiency score; FDR, false discovery rate; HEOG, horizontal electrooculogram; ICA, independent component analysis; IEM, inverted encoding model; MI, modulation index; MVPA, multivariate pattern analysis; PD, positive distractor; RT, reaction time; VEOG, vertical electrooculogram.

for limited attentional resources is likely to be inhibited in advance via a "proactive suppression" mechanism [2]. The emerging consensus on the mechanism of proactive suppression is flexible and not unitary [3,4]; it might be influenced by contextual factors [5,6], statistical learning [7,8], and nonspatial features [9,10]. However, the jury is still out on the mechanism supporting the proactive suppression of spatial distracting information.

Prior behavioral work provided mixed evidence for proactive suppression by considering response time or accuracy. Several studies have shown that providing the distractor-related location in advance is likely to harm [11,12], not influence [13], or benefit the target response [14]. Some behavioral research shows that proactive suppression might not occur unless the location of the upcoming distractor becomes predictable in blocked designs or by repeating stimuli [7,15]. It was suggested that behavioral changes (e.g., reaction time, accuracy) are only indirect indicators of distractor suppression, as they reflect the target-related outcome of all processes from stimulus presentation up until response of the task [16]. The spatial proximity of the distractor plays an important role in determining how proactive suppression impacts behavior [17,18]. As the distance between the target and distractor increases, the accuracy of target selection improves monotonically while reaction time decreases monotonically [19–21]. In recent decades, the dependency of performance on the distance between the target and distractor is calculated by the performance as a function of target-distractor distance [7,22,23].

Using an intermodal task, Foxe and colleagues [24] reported enhanced alpha power in cortical areas when visual stimuli were task-irrelevant, thus suggesting that alpha-band activity is associated with distractor suppression. In a working memory task, alpha power was found to increase in anticipation of distractor [25]. Applying a cued spatial attention task, Worden and colleagues [26] found that posterior alpha-band activity was larger contralateral to the ignored compared to the attended visual field. Given the link between spatial attention and alpha power, the allocation of spatial attention can be indexed by modulation in alpha-band power [27]. A substantial body of work has linked alpha-band activity to locally specific neuronal suppression, and several recent studies have focused on the investigation of alpha power in terms of its hemispheric lateralization and spatial selectivity.

Alpha-band activity can be interpreted as a neural signature of distractor inhibition [9,28–30]. Alpha power increased relatively contralateral to the anticipated irrelevant visual input and has been termed the "negative" alpha modulation of distractors [31]. These findings were complemented by a recent study reporting an increase in alpha power contralateral to predicted distractors [30]. Importantly, such alpha power lateralization was observed before the distractor onset on a trial-by-trial basis, consistent with alpha lateralization as proactive suppression of upcoming distracting inputs [32]. Such alpha lateralization reflects the distractor-related suppression by spatial attention, in line with the gating by the distractor inhibition hypothesis [33]. While the hemispheric lateralization gives an indication of spatial specificity, the lateralized activity alone does not necessarily reflect a precise spatial representation [34]. An influential line of research further focuses on the precise spatial selectivity reflected by the alpha activity by applying encoding models (EMs), which track the temporal and spatial dynamics of spatial attention [3,35–38]. The EMs are based on channel tuning functions (CTFs), in which the spatial distribution of the alpha power across electrodes allows for deriving a more selectivity of the attention bias. Indeed, the fact that the spatially distributed alpha activity precisely tracks the position of the target, even in the absence of irrelevant distractors, casts doubt on whether the functional role of alpha oscillations is consistent with the distractor inhibition hypothesis. Several studies suggest that the evidence for alpha power as a distractor inhibition account is limited [39], and as a consequence, it is debated to what extent alpha oscillation can proactively suppress distractors [3,30,40].

Beyond alpha activity, distractors can elicit the positive distractor ($P_D$) event-related potential (ERP) component, which has been proposed to reflect reactive prevention or termination of salient distractors [41]. The decreased amplitude of the $P_D$ is thought to reduce distractor interference in spatial priority maps [23]. The $P_D$ amplitude can be influenced by learned suppression [3], nonspatial suppression [42], and strategy [30]. Recent studies found a reduction of the $P_D$ following a predictable distractor, reflecting a decrease in attentional suppression [16,30,43,44], while there is still a lack of direct evidence on the relationship between anticipatory alpha activity and subsequent distractor-elicited $P_D$ for spatial suppression, albeit the trial-wise magnitude of the pretarget alpha power has been linked to lateral indices of attentional selection in the ERP [30].

To address these unsettled issues, we applied a variant of the Posner paradigm (Fig 1A) by using spatial circular radar-like cues, where given prior spatial information was informative or uninformative (Experiment 1), with further manipulation for the validity of information (Experiment 2), and symbolic alternation (Experiment 3). We hypothesized that cueing the distractor location would influence the distractor spatial proximity effect on behavioral performance and alpha activities. Then, we aimed to investigate cue-induced alpha activity and the $P_D$ elicited by distractors, as well as their interaction. We assume that if proactive suppression of the upcoming distractor is related to alpha activity, the corresponding changes in alpha power with spatial cueing should explain the variance in $P_D$.

## Results

The materials and methods of these experiments are available at the Open Science Framework (https://osf.io/z9rym/).

### Experiment 1

To study the neural mechanisms underlying distractor suppression guided by spatial cues, we performed 2 sessions in Experiment 1 (see Fig 1A). For the valid-cue session, the radar-like cue was fully predictive of the direction in which the subsequent distractor would appear (red represents the distractor; yellow represents the target). We also included an invalid-cue session in which the distractor location was uninformative. Before each session, participants were informed of the cue validity (valid or invalid) and its corresponding spatial probability of the target and distractor, which guided participants to indicate the orientation of the gray line inside the yellow target circle in the search array. Participants were encouraged to make use of the information provided by the predictive cues, which would help them not to get distracted by the salient distractor. The comparison between invalid-cue session and valid-cue session allowed us to assess cueing effect related to distractor.

### Behavior

To examine the spatial proximity of the distractor, the trial was divided into 9 subgroups according to the relative distances of the target to the distractor location (abbreviated as DTDs). Then, the performance of each subgroup was averaged to examine the response to the target when the distractor appeared at different DTDs (Fig 1D). To quantify the extent of the spatial proximity of the distractor, the slope was characterized by collapsing trials across the same DTD and fitting these data by a linear function (Default function of MATLAB: polyfit. m). Both accuracy (ACC) and reaction time (RT) showed a significant slope in each session (ps < 0.031, two-tailed). To take multivariable behavioral data into account, we further used efficiency scores (ESs) by dividing the ACC by RT [45], which is a standard way to combine ACC and RT measures of performance. A larger ES means more efficient responses. The slope

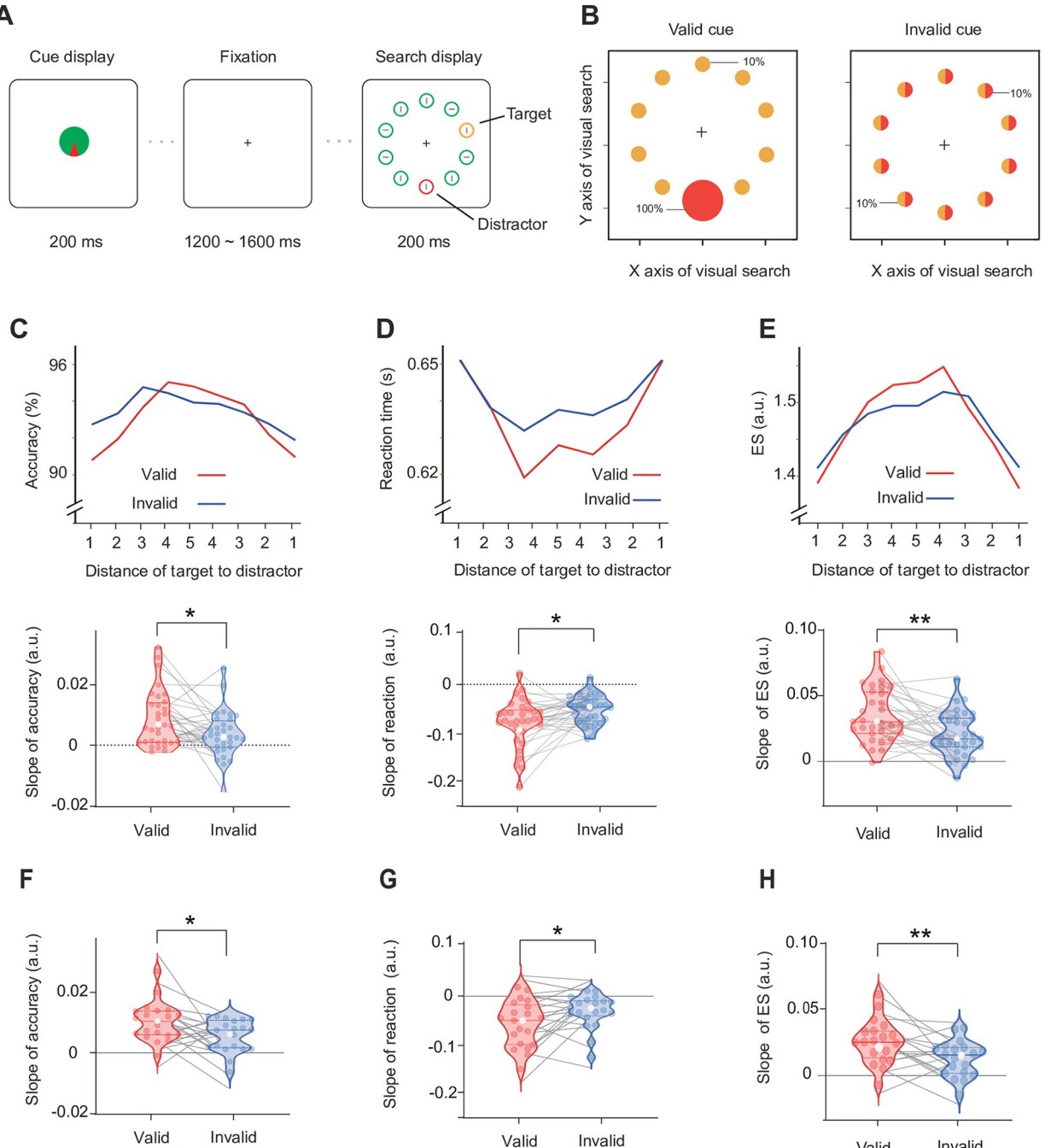

**Fig 1. Task paradigm and behavioral results for Experiment 1. (A)** Each trial began with a cue display of the distractor, 1,200 to 1,600 ms followed by a search display. In 2 separate sessions, the cue display was fully predictive (with 100% validity) or not predictive (with 10% validity) of the specific location of the red distractor circle. Participants were instructed to indicate the orientation of the gray line inside the yellow target circle in the search array. **(B)** The spatial probability of the target and distractor occurring during subsequent visual search with respect to 2 cue sessions (yellow represents the target; red represents the distractor). Before each session, participants were informed of the cue validity (valid or invalid) and its corresponding spatial probability of the target and distractor. **(C)** The mean (top) and slope (bottom) of accuracy across DTD in the valid- (red) and invalid- (blue) cue sessions for Experiment 1. **(D)** The mean (top) and slope (bottom) of reaction time for Experiment 1. **(E)** The mean (top) and slope (bottom) of efficiency scores for Experiment 1. The slope of accuracy **(F)**, reaction times **(G)**, efficiency scores **(H)** in the valid- (red) and invalid- (blue) cue sessions for the behavioral control experiment. Violin plots depict the distributions of measurements in each session, with dots representing each subject. The solid and dotted lines indicate medians and quartiles, respectively. $^{**}p < 0.01$, $^*p < 0.05$ (see S1 Data for raw values).

of ES was significantly larger than zero for the valid-cue session ($t_{29}$ = 9.699, $p$ < 0.001, two-tailed, Cohen's d = 1.715) as well as for the invalid-cue session ($t_{29}$ = 6.799, $p$ < 0.001, two-tailed, Cohen's d = 1.202). As expected, these results are consistent with previous studies [13,14,16,30,34,46] and provide clear evidence for a spatial gradient of suppression surrounding salient distractors.

A two-way repeated ANOVA with session (valid, invalid) and DTD (first, second, third, fourth, and fifth) as factors was conducted on mean ESs. The significant main effect of DTD ($F_{4,\ 116}$ = 12.177, $p$ < 0.001, $\eta^2$ = 0.925) suggested that the magnitude of ESs depended on the target-distractor distance, which is consistent with previous studies [23,39]. As expected, we also found a significant session-by-DTD interaction ($F_{4,\ 116}$ = 3.471, $p$ = 0.037, $\eta^2$ = 0.314). We suggest that cueing distractor explicitly would influence the distractor spatial proximity effect on behavioral performance. Compared with the invalid-cue session, we found that participants had better performance when distractors occurred at locations (the fifth and fourth) far away from the target in the valid-cue session. In contrast, participants had poorer performance when distractors occurred at locations (first and second) near the target. If the distractor cueing effect has a spatial extent, we expect that the slope of ES in the valid-cue session may be steeper than that for the invalid-cue session. As expected, the significant cueing effects were found on both the slope of ACC ($t_{29}$ = 2.556, $p$ = 0.016, two-tailed, Cohen's d = 0.452; Fig 1C) and RT ($t_{29}$ = −2.579, $p$ = 0.015, two-tailed, Cohen's d = −0.456; Fig 1D). We obtained a similar cueing effect in participants' the slope of ES ($t_{29}$ = 3.356, $p$ = 0.002, two-tailed, Cohen's d = 0.593; Fig 1E) in Experiment 1. Taken together, our preliminary results showed novel spatial behavioral changes, which supported the existence of proactive suppression for spatial distractor cues. We also analyzed the mean performance for Experiment 1. Our behavioral results (Fig B in S1 Appendix) were consistent with previous studies [4,7,16] that found no difference in reaction time and accuracy between validly and invalidly cued sessions. Please see S1 Appendix for details.

To provide general behavioral evidence and exclude the possible influence of the red color, we conducted a behavioral control experiment by adding distractor-absent trials and changed half of the blocks to blue distractors instead of red distractors. We aimed to investigate whether a valid spatial cue can decrease the distractor effect. In the behavioral control experiment, we still analyzed the same slope of ACC, RT, and ES as in Experiment 1 to examine spatial changes in behavioral outcomes. We also found a significant distractor cueing effect on the slope of ACC ($t_{20}$ = 2.398, $p$ = 0.026, two-tailed, Cohen's d = 0.523; Fig 1F), RT ($t_{20}$ = −2.451, $p$ = 0.024, two-tailed, Cohen's d = −0.535; Fig 1G), and ES ($t_{20}$ = 2.778, $p$ = 0.012, two-tailed, Cohen's d = 0.606; Fig 1H). Similar to Experiment 1, no significant distractor cueing effects (valid minus invalid) of mean ACC, RT, or ES were found ($p$ > 0.371). These results repeated the findings in Experiment 1, suggesting that the cueing distractor explicitly truly influenced the distractor spatial proximity effect on behavioral performance, but not in relation to the specific color. We also analyzed the distractor capture effect and suggested that a distractor could be proactively inhibited when a spatial cue was presented that indicated the location of the distractor (see S1 Appendix).

## Alpha channel tuning function (CTF) of distractor cueing

Previous research suggested that the spatial distribution of neural representation was especially pronounced within the alpha band power (8 to 12 Hz). The inverted encoding model (IEM) analysis was applied to reconstruct the inhibited distractor location from the pattern of alpha power to obtain a high-resolution spatiotemporal profile. As shown in Fig 2A, this procedure produces CTFs, which reflect the spatial distribution of alpha power that is measured by scalp

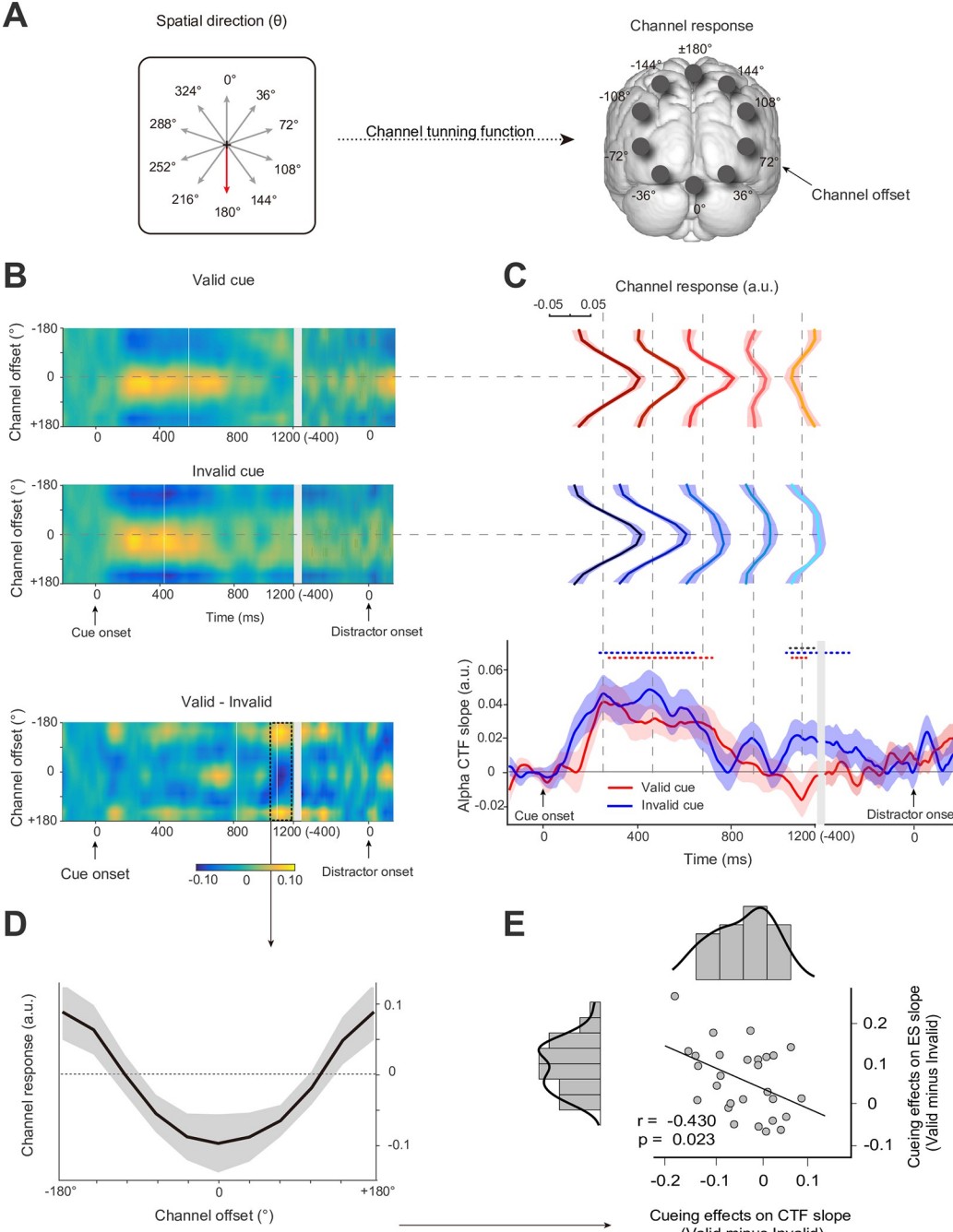

**Fig 2. EEG results during the cue-distractor intervals from Experiment 1. (A)** The spatial direction of the distractor cue varied from trial to trial. The spatial distribution of alpha power was modeled by the channel tuning functions (CTFs) across 10 ideal channel offsets; right panel shows channel offsets and the centre channel if distractor cue point 180 degrees (red arrow). **(B)** Alpha-band CTFs across the cue-distractor intervals for valid-cue and invalid-cue sessions. The difference between the 2 sessions was also plotted. **(C)** The direction selectivity of the alpha-band CTF (measured as CTF slope) across time in valid- (red) and invalid- (blue) cue sessions. The different channel response curves at 5 sampled time points (gray vertical dashed lines) were plotted in both sessions. The red and blue dashed lines at the top indicate clusters where sessions differed significantly from zero ($p < 0.05$), and significant differences between sessions are marked by the black dashed line ($p < 0.05$). **(D)** The cueing effect on alpha-band CTFs (valid−invalid; averaged from 1,040 to 1,200 ms) is related to anticipation of the distractor. **(E)** Correlation between alpha CTF slope and efficiency scores (ESs) slope. The ES difference between valid and invalid cues (cueing effects) varies as a function of the alpha CTF slope difference between valid and invalid cues. Positive values of the y-axis indicate larger ES slope with valid cues than with invalid cues. Negative values of the x-axis indicate that the negative-going alpha CTF slope was larger with valid cues than with invalid cues (see S2 Data for raw values).

EEG (conceptualized into 10 ideal electrodes). In brief, the center channel was tuned for the position of the direction of interest (e.g., 180˚ red arrow in Fig 2A, left), then channel offsets (e.g., 72˚ in Fig 2A, right) were defined as the angular difference between the center channel and other channels. Each estimated CTF was then circularly shifted to a common center (0˚ on the channel offset axes of Fig 2B) and several channel offsets (−180˚ to 180˚). The final CTF was a function associated with the shifted channel offsets.

Fig 2B shows CTFs across the cue-distractor intervals (cue-locked: −200 to 1,200 ms; distractor-locked: −400 to 200 ms) for valid- and invalid-cue sessions. To measure the spatial selectivity of channel responses, the time-resolved slope of CTFs was calculated for both the valid- (red lines in Fig 2C) and invalid-cue sessions (blue lines in Fig 2C). Channel response curves were plotted at different sampled time points from the maximum ($T_1$: 224 ms) to the minimum ($T_5$: 1,136 ms) of the slope of CTFs for the valid-cue session and a set of equal diversion points between $T_1$ and $T_5$ ($T_2$: 452 ms, $T_3$: 680 ms, $T_4$: 908 ms). These results suggest that CTFs are sensitive to the inhibited distractor location and time course, which was tracked by the spatial response of alpha power across different channel offsets.

In both valid- and invalid-cue sessions, the distractor selectivity (positive slope of CTFs) shows an initial steep rise followed by a gradual decrease, resulting in a slope significantly (cluster-based permutation test: $p < 0.050$, two-tailed) different from zero in valid-cue sessions (from 264 to 744 ms) and invalid-cue sessions (from 196 to 648 ms). This shows that the alpha power was first selective for cued direction regardless of whether it had distractor-related information. Then, invalid cues still led to a significant slope from 1,064 to 1,200 ms locked to cue display and from −400 to −236 ms locked to the distractor (permutation test: $p < 0.050$, two-tailed), suggesting that channels continued to be selective for the cued location in invalid-cue sessions.

Given that the positive slope of CTFs represents the selectivity of neural activity responses to cued distractor location, the negative slope may represent the suppression of neural activity responses to the distractor. In contrast, valid cues led to distractor suppression (negative slope of CTFs) from 1,062 to 1,168 ms (Fig 2C, bottom panel; permutation test: $p < 0.050$, two-tailed), resulting in a significant distractor cueing effect on the slope of CTFs from 1,040 to 1,200 ms between the valid-cue session and the invalid-cue session (permutation test: $p < 0.050$, two-tailed). The mean difference in CTFs (valid−invalid) in the significant time windows was averaged to identify the change in the channel response curve. As shown in Fig 2D, channel response relatively increased at channels contralateral to the cued distractor location, and channel response relatively decreased at channels ipsilateral to the cued distractor location. Furthermore, we investigated whether the cueing effects on ES slope were correlated with the cueing effects on the alpha CTF slope. A negative correlation (r = −0.430, $p = 0.023$; two-tailed) indicated that the larger the difference in the negative-going alpha CTF slope between valid cues and invalid cues, the larger the cueing effects on the ES slope (Fig 2E). We suggested that the spatial distribution of the alpha power response for an upcoming distractor can influence the distractor spatial proximity effect on behavioral performance.

Together, our results show dynamic spatial alpha power tuning to the cued distractor location during the cue-distractor interval. The negative CTF slope was only observed in valid-cue sessions, which indicates that cueing distractors might suppress spatially subsequent distracting input by flipping the spatial tuning to the inhibited distractor location in advance.

## Alpha MI of distractor cueing

Then, our interest lay in specific spatial distribution effects of alpha power—lateralized alpha power. This lateralized alpha power is defined as the difference between the alpha power in the

contralateral hemisphere and that in the ipsilateral hemisphere with respect to distractor and is usually measured by the alpha modulation index (MI) [47,48]. To enable isolation of lateralized distractor-specific alpha power, the alpha MI evoked by cues was computed based on trials where the cue point was 4 of 10 possible directions (288˚, 108˚, 252˚, 72˚). Trials were categorized as left-cued when they pointed 288˚ or 252˚, whereas those that pointed 108˚ or 72˚ were classified as right-cued trials. Then, we combined alpha band power for left-cued trials minus right-cued trials, normalized by their mean, and averaged over left and right (see Materials and methods for details).

As shown in the time-course representation in Fig 3A, mimicking the CTF findings, our results show that the amplitude of MI was significantly positively modulated during the 244 to 560 ms and 1,012 to 1,200 ms invalid-cue sessions (permutation test: $p < 0.050$, two-tailed). The alpha MI in valid-cue sessions showed a significant positive modulation from 208 to 348 ms and a significant negative modulation during the late period from 748 to 1,012 ms (permutation test: $p < 0.050$, two-tailed). Testing for distractor cueing effects revealed a significant difference between valid- and invalid-cue sessions during the late period of 888 to 1,200 ms ($p < 0.050$, two-tailed). This result suggested that for the valid cue, the alpha power was more strongly elevated over the hemisphere contralateral to the cued distractor field during later stages.

### Distractor-elicited ERP

We then focused on the ERPs during the subsequent visual search display. We only used trials with a lateral distractor and midline target present, in which a lateral distractor can evoke $P_D$ components. The $P_D$ component was present as a positive deflection in the ERP waveform at the visual cortex contralateral relative to ipsilateral to the distractor. Fig 3C showed the difference waveforms (contralateral minus ipsilateral), revealing that a significant positive difference (248 to 316 ms; cluster-based permutation test: $p < 0.050$; Fig 3C) in the invalid-cue sessions and a significant cueing effect on difference waveforms was apparent at P7/8 electrodes (258 to 302 ms; cluster-based permutation test: $p < 0.050$; Fig 3C). The amplitude of $P_D$ was significant in the invalid-cue session ($t_{29} = 2.228$, $p = 0.034$, two-tailed, Cohen's d = 0.414) but not in the valid-cue session ($t_{29} = -0.007$, $p = 0.995$, two-tailed, Cohen's d = −0.001), which resulted in a significant $P_D$ difference between the 2 sessions ($t_{29} = -2.090$, $p = 0.046$, two-tailed, Cohen's d = −0.388). These results were consistent with prior work reporting distractor reductions in $P_D$ amplitude [3,16,30,43]. Our results suggested that cueing distractors appeared to be a reduced need to reactively inhibit the capture of salience distractors, as evidenced by reduced $P_D$ amplitude. As shown in Fig 3C, the $P_D$ is preceded by significant contralateral negativity in the valid-cue session (172 to 236 ms) and in invalid-cue session (192 to 218 ms; cluster-based permutation test: $p < 0.050$, two-tailed), which is called the distractor-elicited N2pc in previous studies [7,30]. No significant difference in distractor-elicited N2pc between valid- and invalid-cue sessions was found ($p = 0.225$).

We also conducted decoding analyses with ERP waveforms across all electrodes. The decoding result reveals that when the distractor could be predicted, participants had a better spatial representation of the target, which supports the cueing distractor benefitting the spatial representation for target selection during the search array (see S1 Appendix for details).

### Correlation analysis between behavioral performance and electrophysiological signals

We further investigated whether the behavioral performance was correlated with the cue-induced alpha activity or distractor-elicited $P_D$. For reaction time, accuracy, mean ES, and

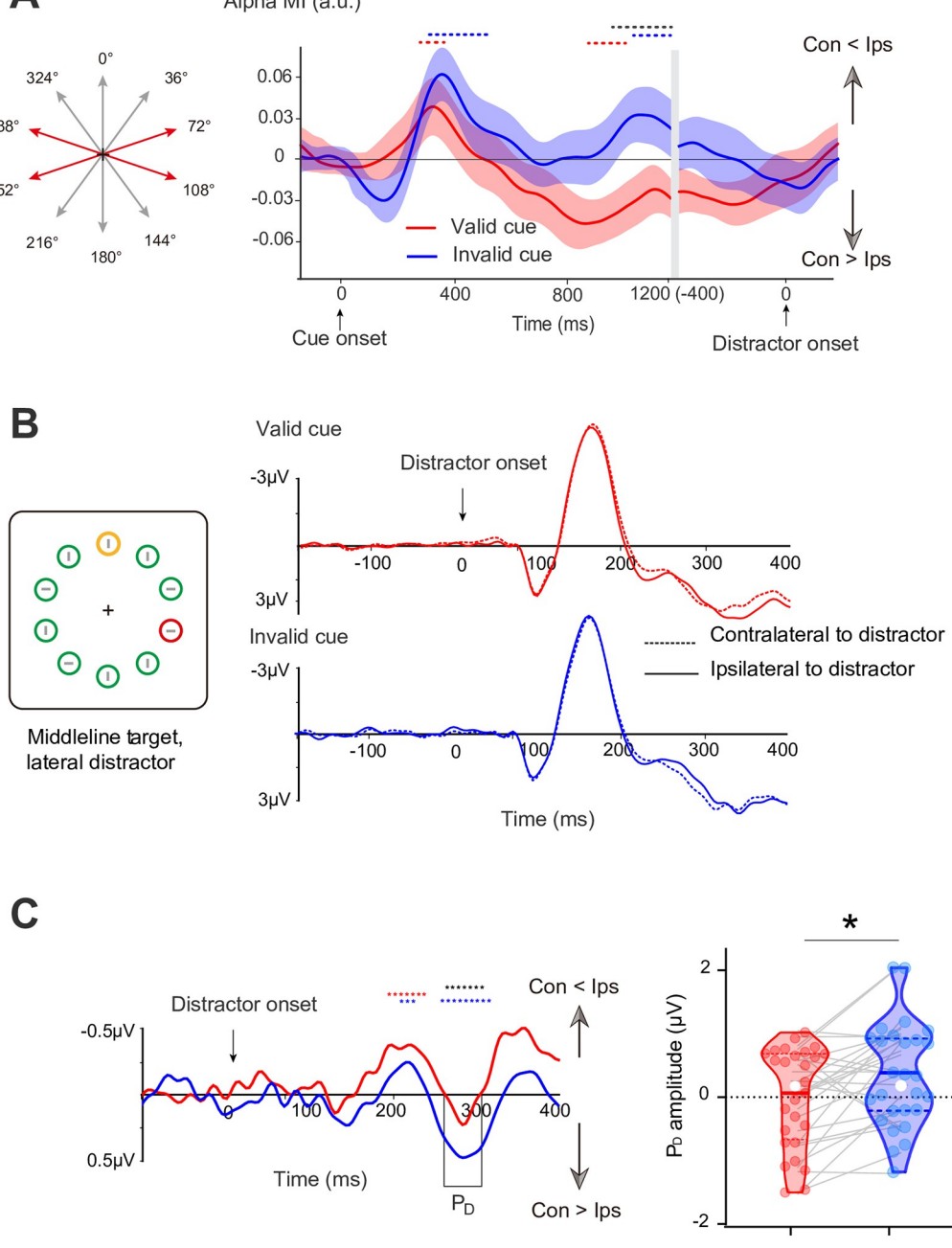

**Fig 3. Lateralized EEG results during the stimulus period from Experiment 1. (A)** Time course of the alpha modulation index in the posterior electrodes for valid- (red) and invalid- (blue) cue sessions. The red and blue dashed lines indicate a significant difference from zero, and the black dashed line indicates clusters with a significant difference between 2 sessions ($p < 0.05$). Shades of light color along with the dark color lines represent error bars (±1 SEM). **(B)** Grand averaged ERPs at contralateral and ipsilateral electrode sites relative to the distractor (averaged over P7 and P8) in valid- (red) and invalid- (blue) cue sessions. **(C)** The left panel shows the corresponding difference waves (contralateral minus ipsilateral activity) for valid- (red lines) and invalid-cue sessions (blue lines). Violin plots depict the $P_D$ amplitude (248- to 316 ms) in the 2 sessions, with the dots representing each subject. The solid and dotted lines indicate medians and quartiles, respectively. *$p < 0.05$. Con, contralateral to distractor cue; Ips, ipsilateral to distractor cue (see S3 Data for raw values).

slope of ES, no correlation was found at the between-subject level (ps > 0.341). At the single-trial level, no significant difference was found among the quartiles for reaction time and accuracy (ps > 0.120). Interestingly, the cueing effects on the ES slope were correlated with the cueing effects on the alpha CTF slope. A negative correlation (r = −0.430, $p$ = 0.023; two-tailed) indicated that the larger the negative-going alpha CTF slope with valid cues relative to invalid cues, the larger the cueing effects on ES slope (Fig 2E). We suggested that the spatial distribution of alpha power response for an upcoming distractor influenced the spatial proximity of the distractor.

## Experiment 2

In Experiment 2 (see Fig 4A), we manipulated the spatial probability of the target and distractor occurring during subsequent visual search respectively. Cue pointed left or right and informed the participants of the approximate scope in which the upcoming distractor would occur in the search display, instead of the exact location in Experiment 1. The variable scope of the distractor cue across 3 trials was related to the predictive validity of distractor occurrence. On the one hand, this manipulation of predictive validity allowed us to exclude the possibility that the hypothesized evidence for proactive suppression in Experiment 1 simply reflects the information gap between the informative cue (valid) and uninformative cue (invalid). On the other hand, distractor cueing explicitly removed a potential target location in Experiment 1, increasing the probability that the target would appear in the contralateral hemifield. This potentially confounds the lateral effect related to suppression with lateral effects associated with the deployment of attention to the contralateral field. Here, to control the occurrence of a subsequent target location, we pseudorandomized the location of the target circle by specifying a uniform spatial probability of 4.25% on each lateral location and 33% on each midline location (see Fig 4A; right panel). This manipulation of the target appearing more often on the vertical meridian allowed us to isolate the lateralized brain activity related to distractor anticipation, which relies on the fact that stimuli on the vertical meridian target do not elicit the lateralized activities [49,50]. Note that the target had exactly the same spatial probability across the different distractor predictive validity conditions, and the location of the subsequent target was totally independent of the distractor location. Thus, participants have no incentive to use distractor cues to infer the probability of a location containing a target.

## Behavior

Experiment 2 adopted the methods and indicators of Experiment 1 for behavioral analysis. Note that spatial probabilities of the target were not uniform across spatial locations but remained equivalent among all trials (see Fig 4A), which allowed us to compare responses for the target with variable spatial probabilities of a distractor. As in Experiment 2, behavioral results (Fig 4B) showed a main effect of predictive validity on slope of ES ($F_{2, 50}$ = 34.480, $p$ < 0.001, $\eta^2$ = 0.579). This finding suggests that cueing distractors with variable-predictive validity influences on the subsequent distractor spatial proximity effect on behavioral performance. Planned pairwise comparisons for the slope of ES (Fig 4B) again showed a prominent cueing effect in high-predictive validity trials (high minus null: $t_{25}$ = 7.192, $p$ < 0.001, two-tailed, Cohen's d = 1.411) and low-predictive validity trials (low minus null: $t_{25}$ = 6.703, $p$ < 0.001, two-tailed, Cohen's d = 1.315). However, no significant difference between high- and low-predictive validity trails was observed for slope of ES ($t_{25}$ = 1.171, $p$ = 0.252, two-tailed, Cohen's d = 0.230). These results may be due to a ceiling effect of behavioral responses or limitations of current testing paradigms for examining "responses for target" in distractor-related manipulation. We also did analysis mean performance for Experiment 2; please see S1 Appendix for details.

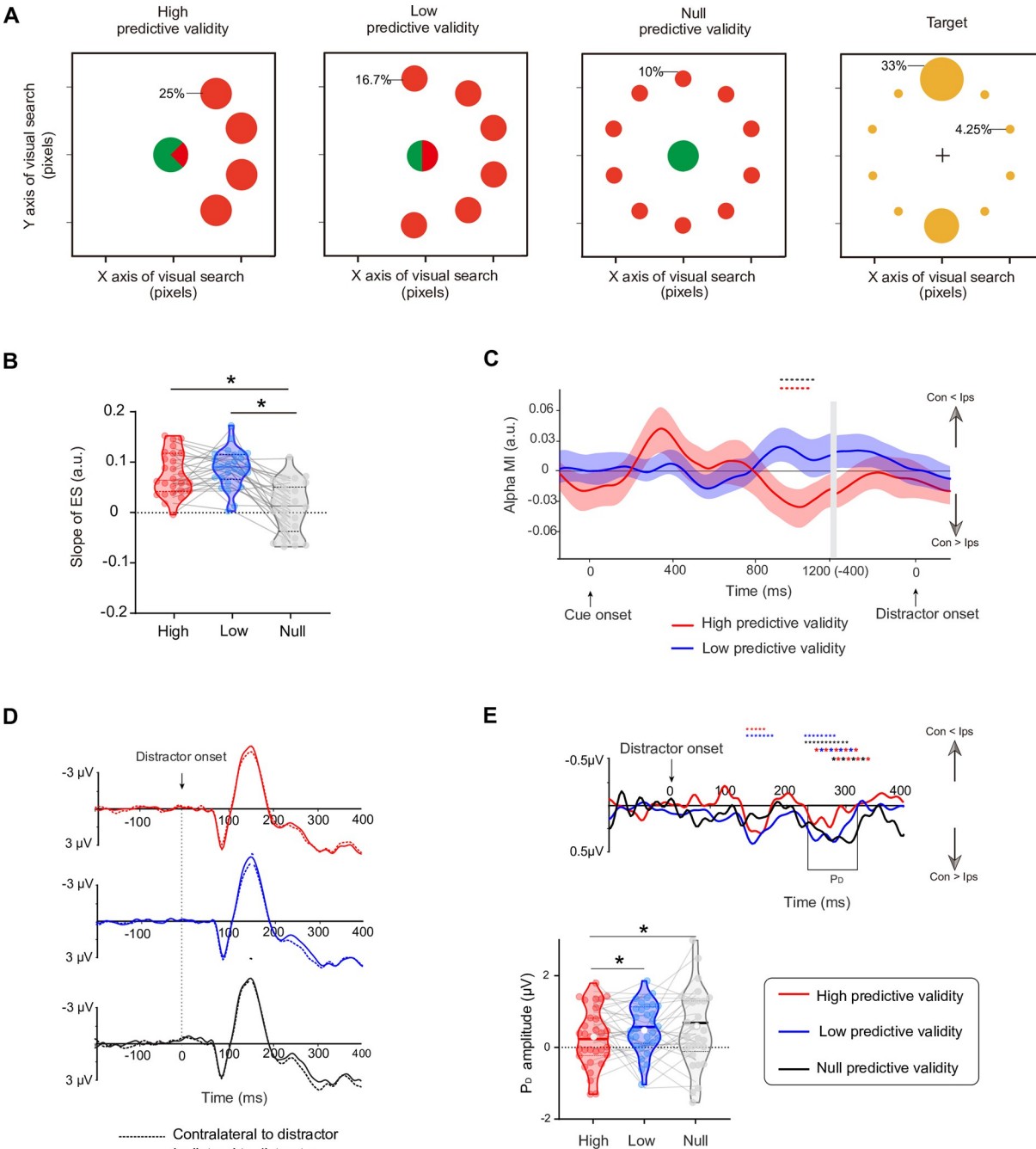

**Fig 4. Task paradigm and EEG results for Experiment 2. (A)** Three types of cue displays and corresponding spatial probability of the target and distractor occurring during subsequent visual search. Note that spatial probability was conceptual and did not actually appear around the cue. **(B)** The slope of ES for high- (red), low- (blue), and null- (black) predictive validity trials. **(C)** Time course of the alpha MI in the posterior electrodes for high- (red) and low- (blue) predictive validity trials. Shades of light color along with the dark color lines represent error bars (±1 SEM). **(D)** Grand averaged ERPs at contralateral and ipsilateral electrode sites relative to the distractor (averaged over P7 and P8) in high- (red) and low- (blue) predictive validity trials. **(E)** The upper panel shows the corresponding difference waves (contralateral minus ipsilateral activity) for high- (red lines), low- (blue lines), and null-predictive validity. The red, blue, and black dashed lines indicate a significant difference from zero, and the dashed lines with 2 colors indicate clusters with a significant difference between the 2 conditions ($p < 0.05$). The lower panel shows scatter plot for $P_D$ amplitude, with the dots representing each subject. $^*p < 0.05$. Con, contralateral to distractor cue; Ips, ipsilateral to distractor cue (see S4 Data for raw values).

## Alpha MI of distractor cueing

The alpha MI of high-predictive validity (red line) and low-predictive validity (blue line) trials during the cue period are shown in Fig 4C. Due to the cue display without lateralized spatial information, we did not analyze alpha MI in null-predictive validity trials. Our results showed that a significant negative alpha MI occurred only in high-predictive validity trials during the late period of 878 to 1,148 ms, which suggested the alpha power increased in the contralateral hemisphere to distractor cue (cluster-based permutation test: $p < 0.050$, two-tailed). Post hoc analysis revealed that alpha MI in high-predictive validity trials was significantly lower than that in low-predictive validity trials (high minus low: 867 to 1,113 ms; permutation test: $p < 0.050$, two-tailed).

## Distractor-elicited ERP

We anticipated that as the predictive validity of the distractor cue increased, the participant's reactive suppression of the subsequent salient distractor in the search array would decrease, resulting in a smaller distractor-elicited $P_D$. Our results showed that significant $P_D$ (cluster-based permutation test: $p < 0.050$) was apparent at P7/8 electrodes in the low-predictive trials (218 to 296 ms; $t_{25} = 3.418$, $p = 0.004$, two-tailed, Cohen's d = 0.514) and null-predictive trials (210 to 312 ms; $t_{25} = 2.18$, $p = 0.044$, two-tailed, Cohen's d = 0.311), but not in the valid-cue trials ($t_{25} = -0.178$, $p = 0.691$, two-tailed, Cohen's d = −0.017). As expected, the results in Fig 4D showed a significant main effect of predictive validity on $P_D$ ($F_{2, 50} = 3.173$, $p = 0.049$, $\eta^2 = 0.099$). Further paired $t$ tests confirmed that the $P_D$ elicited by expected distractors in high-predictive validity trials was greatly reduced in amplitude compared to expected distractors in low-predictive validity trials ($t_{25} = -2.126$, $p = 0.042$, Cohen's d = −0.388) and unexpected distractors in null-predictive validity trials ($t_{25} = -2.266$, $p = 0.031$, Cohen's d = −0.414). The distractor-elicited $P_D$ did not differ between low- and null-predictive validity trials (ps > 0.250, $BF_{10} < 0.333$).

As shown in Fig 4, the $P_D$ is preceded by significant contralateral positivity in both the high- (152 to 186 ms) and the low-predictive validity trial (150 to 198 ms; cluster-based permutation test: $p < 0.050$, two-tailed), which is called early $P_D$ in previous studies [30,51]. No significant difference in early $P_D$ between null- and other predictive validity conditions was found (ps > 0.458, $BF_{10} < 0.333$).

## Correlation analysis between alpha modulation and distractor-elicited $P_D$

We used correlation analysis to investigate the relationship between cue-induced alpha lateralization and subsequent distractor-elicited $P_D$ in high- and low-predictive validity trials. When predictive validity of the distractor cue was high, we found a significant correlation between negative alpha MI (averaged 800 to 1,100 ms) and $P_D$ amplitude (Fig 5A, left; r = 0.410, $p = 0.041$), which suggested that subjects with more alpha power contralateral to the cued distractor (negative alpha MI) during the cue-distractor period showed smaller distractor-elicited $P_D$ amplitude in subsequent visual searches. However, there was no significant correlation between alpha MI and $P_D$ amplitude (Fig 5C, right; r = 0.025, $p = 0.909$) when the predictive validity of the cueing distractor was relatively low.

Between-subject correlation analysis is sensitive to a third variable, such as better electrode contacts, less muscle noise, or more motivation. Several studies have investigated the alpha power at the within-subjects level. van Dijk and colleagues [52] found that the trial-wise variance of pretarget lateral alpha power is related to visual discrimination ability. van Zoest and colleagues [30] further found that trial-wise variance of pretarget alpha power is related to the distractor-elicited N2pc and early $P_D$. Hence, we further calculated the average single-trial $P_D$

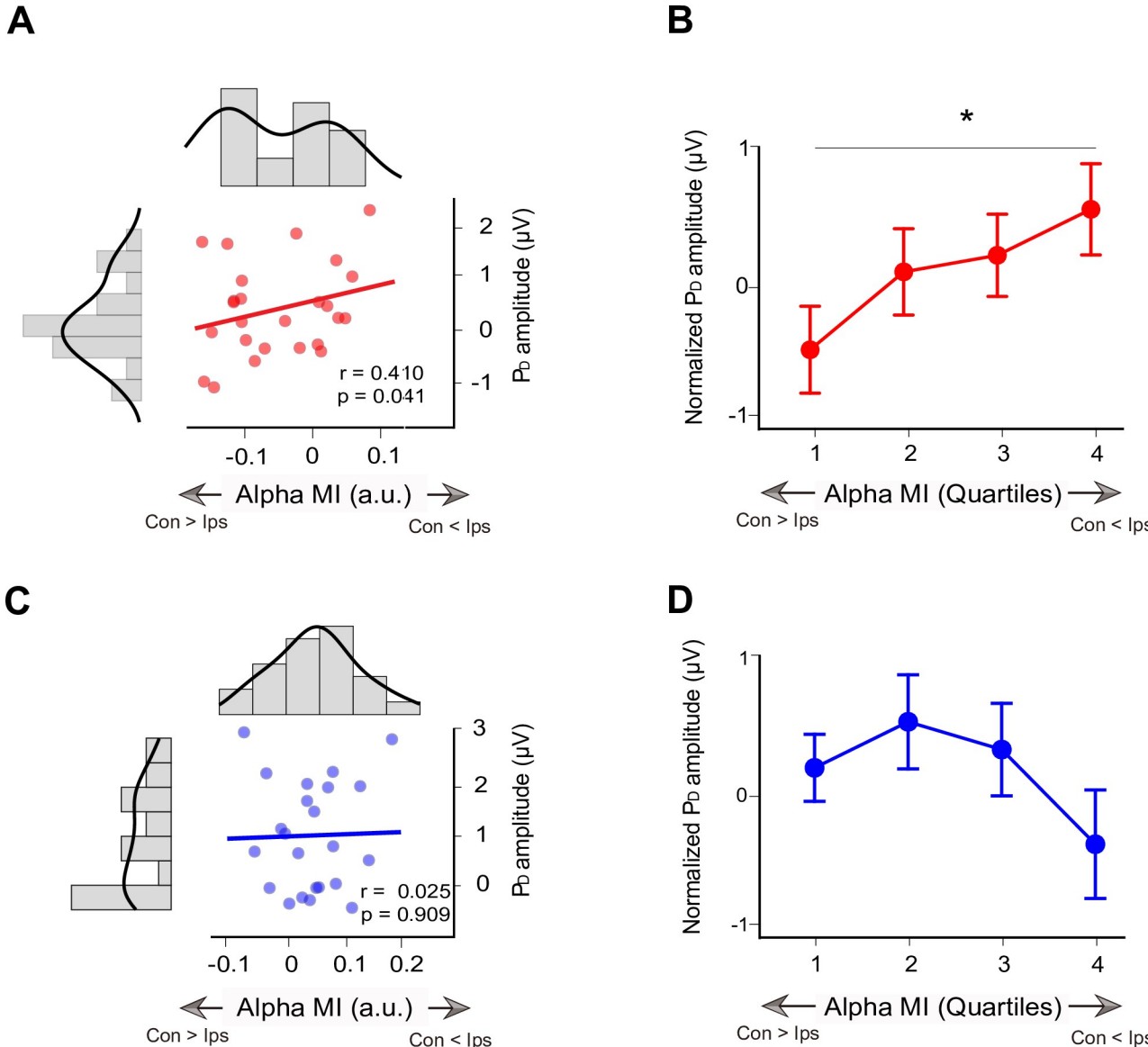

**Fig 5. Relationship between alpha MI and $P_D$ in Experiment 2. (A)** Alpha MI during the cue period as a function of the subsequent distractor-elicited $P_D$ amplitudes during a visual search between participants in high-predictive validity trials. The diagrams along with the scatter plot are the frequency distributions of alpha MI and $P_D$ amplitude, respectively. **(B)** Averaged single-trial $P_D$ for each quartile at the within-subjects level in high-predictive validity trials. The trials were sorted according to cue-induced alpha MI and binned into quartiles. The $P_D$ amplitudes were normalized and then averaged over subjects. $^* p < 0.05$. **(C)** Scatter plot for low-predictive validity trials. **(D)** Quartile plot for low-predictive validity trials. Con, contralateral to distractor cue; Ips, ipsilateral to distractor cue (see S5 Data for raw values).

for each quartile to confirm the relationship between the alpha MI and subsequent $P_D$ amplitude. Although the repeated-measures ANOVA of $P_D$ did not reach significance ($F_{3, 75}$ = 1.818, $p = 0.152$), the results in the high-predictive validity trials show that the $P_D$ amplitude in the fourth negative quartile was significantly larger than that in the first negative quartile (Fig 5B, left; $t_{25}$ = 2.303, $p = 0.030$, Cohen's d = 0.461). Accordingly, we suggested that the trials with more alpha power contralateral to the cued distractor (negative alpha MI) also showed smaller distractor-elicited $P_D$ amplitude subsequently. Similarly, no significant difference was found among the quartiles (Fig 5D, right; ps > 0.157) in the low-predictive validity trials.

These results showed that there was a close relationship between alpha MI and subsequent bio-markers of distractor suppression at both the between- and within-subjects levels when the predictive validity of distractor cues was high.

## Correlation analysis between behavioral performance and electrophysiological signals

We further investigated whether behavioral performance was correlated with the cue-induced alpha MI or distractor-elicited $P_D$. For reaction time, accuracy, mean ES, and slope of ES, no correlation was found at the between-subject level (ps > 0.161). At the single-trial level, no significant difference was found among the quartiles for reaction time and accuracy (ps > 0.330).

## Experiment 3

To date, the evidence for distractor processes was confined to tasks with a graphic cue and a modest sample size. On the one hand, previous works have shown that varying shapes of cues might have different impacts on the allocation of spatial covert attention [53,54]. In particular, radar might have different levels of distraction due to the referential position of the subjects with respect to the main scene [54]. It is unclear whether the results from Experiments 1 and 2 were specific to circular radar-like cues or more generally applicable. On the other hand, a modest sample size can increase the false-positive rate and give rise to inflated effect sizes [55]. Thus, the purpose of Experiment 3 was 2-fold: (1) further investigate the alpha power modulation of the distractor cue by using the arrow cue to rule out any graph-specific effects and (2) to explore the potential relationship between distractor anticipation and subsequent distractor inhibition based on large sample size ($N > 40$). The same analysis pipeline as Experiment 2 was applied in Experiment 3.

As shown in Fig 6A, we again isolated significant negative alpha MI (8 to 12 Hz) for distractor cues during late cue-distractor intervals (cluster-based permutation test: $p < 0.050$, two-tailed). Grand averaged ERPs locked to distractor onset were generated to calculate the $P_D$ component. The $P_D$ was significantly different than zero from 234 to 330 ms (Fig 6C, cluster-based permutation test: $p < 0.050$, two-tailed). The scatter plot showed a significant correlation between alpha MI (averaged 750 to 950 ms) and $P_D$ amplitude (Fig 6D, r = 0.332, $p = 0.028$). We suggest that subjects with a more alpha power contralateral to the cued distractor have a lower $P_D$ amplitude in subsequent visual searches.

Here, we also calculated the average single-trial $P_D$ for each quartile at the within-subjects level by the same method as applied in Experiment 2. We found that the alpha MI induced by spatial cues strongly correlated with the subsequent $P_D$ component: The normalized $P_D$ amplitude decreased with an increase in alpha power contralateral to the cued distractor (Fig 6E, repeated-measures ANOVA, $F_{3, 123} = 3.078$, $p = 0.030$). Simple first contrast shows that the $P_D$ amplitude in the fourth negative quartile was significantly larger than that in the first ($t_{41} = 2.059$, $p = 0.046$, Cohen's d = 0.330) and second negative quartiles ($t_{41} = 2.171$, $p = 0.036$, Cohen's d = 0.348). The $P_D$ amplitude in the third negative quartile was significantly higher than that in the first negative quartile ($t_{41} = 2.184$, $p = 0.035$, Cohen's d = 0.350), suggesting that the trials with more alpha power contralateral to the cued distractor during the cue-distractor period have a less distractor-elicited $P_D$ amplitude in the following visual search. In sum, a close relationship between alpha lateralization and subsequent biomarkers of distractor suppression was further confirmed between and within subjects in Experiment 3. We further investigated whether the slope of ES is correlated with the alpha MI or the amplitude of $P_D$ in Experiment 3, no correlation was found at the between-subject level ($p = 0.331$) and the within-subject level ($p = 0.612$).

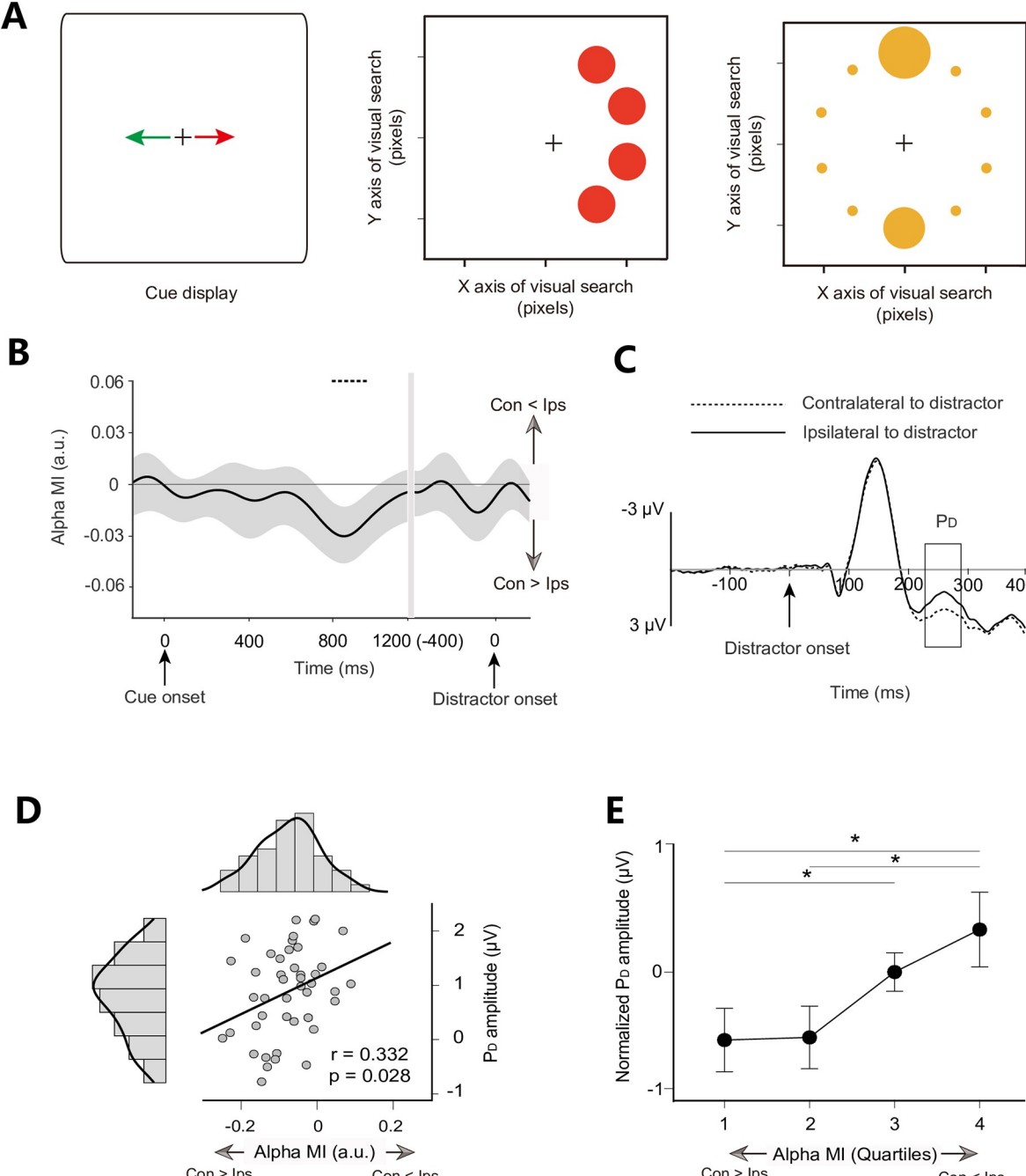

**Fig 6. Task paradigm and EEG results for Experiment 3. (A)** The arrow was fully predictive of the side on which the distractor circle of the corresponding color would subsequently appear. **(B)** Time course of the alpha MI. **(C)** Grand averaged ERPs at contralateral and ipsilateral electrode sites relative to the distractor. **(D)** The scatter plot between cue-induced alpha MI (averaged over the time-frequency windows highlighted by black outlines) and distractor-elicited P_D amplitudes between participants showed a significant correlation. The diagrams along with the scatter plot are the frequency distributions of alpha MI and P_D amplitude, respectively. **(E)** Averaged single-trial P_D for each quartile at the within-subjects level. Trials were sorted according to cue-induced alpha MI and binned into quartiles. P_D amplitudes were normalized and then averaged over subjects. $^*p < 0.05$. Con, contralateral to distractor cue; Ips, ipsilateral to distractor cue (see S6 Data for raw values).

## Discussion

The current study provide insight into the neural mechanisms underlying proactive suppression guided by spatial cues. Across 3 experiments, we present evidence on the existence of alpha activity related to proactive suppression and how it shapes subsequent distractor processing. In Experiment 1, a cueing distractor could sharpen the spatial behavioral measurement (slope of ES), induce distractor suppression (negative alpha CTF slope), and reduce distractor interference ($P_D$ amplitude). The analysis further showed that negative alpha CTF was related to reduced distractor-elicited $P_D$. Results from Experiment 2 demonstrated that increased alpha power contralateral to the distractor (negative alpha MI) and the reduced $P_D$ may be the result of distractor suppression derived from spatial effectiveness. Crucially, when spatial cues with high-predictive validity were employed, the $P_D$ amplitude was observed as a function of cue-elicited alpha MI. That is, increased alpha power contralateral to the cued distractor (negative alpha MI) resulted in less distractor interference, as reflected in the decreased $P_D$. Additionally, a symbolic cue with high-predictive validity was further employed in Experiment 3. We also demonstrated a significant correlation between anticipatory alpha activity and subsequent distractor-elicited $P_D$ across individuals and within individuals. Together, these results have shown how spatial distractor foreknowledge proactively reduces distractor interference.

Behaviorally, we found changes of the cueing effect with different spatial proximity of the distractor in Experiment 1: Cueing distractors precisely had a trend towards harming the performance on the target that appeared in close spatial proximity of the distractor, but boosting the performance on the target appeared in faraway spatial proximity of the distractor. In control behavior experiment and Experiment 2, we also found such spatial proximity of the distractor. The relationship between CTF slope and ES slope provides a neuronal account for the behavioral phenomena that the spatial distribution of alpha power response for upcoming distractors can influence the distractor spatial proximity effect on behavioral performance. With respect to the invalid-cue session, the channel response relatively increased as alpha power decreased at electrodes contralateral to the cued distractor location, where the target was enhanced more easily. Thus, we observed that the cue is more effective when it predicts distractors appearing further away from the target. In contrast, the channel response relatively decreased as alpha power increased at electrodes ipsilateral to the cued distractor location, where the target was suppressed more easily. Thus, the cue is less effective when it predicts distractors appearing closer to the target. A recent study [30] reported that the cue was effective at reducing misdirection of the eyes to the distractor only when the target and distractor were in close spatial proximity, which seemed to be contrary to our behavioral results. We suggested that these inconsistent results might be explained by the large difference in raw latency of eye movements versus manual response. Previous studies have suggested that distractor spatial proximity effect was influenced not only by statistical learning [7], but also by the spatial cues [30]. Our results further extend our current understanding of the distractor spatial proximity effect on behavioral performance and emphasize the importance of anticipatory alpha activities for suppression of distracting inputs.

From distinct research lines, alpha CTF (Fig 2C) and alpha MI (Fig 3A) provided convergent evidence for the dynamic characteristic of proactive suppression as reported in Experiment 1. In the beginning, regardless of task relevance, cues with spatial information may result in increased distractor selectivity (positive alpha CTF slope) or relatively decreased alpha power contralateral to cue distractor (positive alpha MI) at the early stages. This result was consistent with Foster's [35] findings and suggests that enhanced tuning towards cued directions might be first represented in our spatial attention. Then, we observed that our brain

engages the progressive attenuation of both the amplitude of alpha MI and CTF slope. We suggest that this result stems from a white bear metaphor [12], in which participants have to make an effort to minimize interference from the distractor per se when provided the distractor location. Interestingly, during the late preparatory stages, our results clearly show the distractor suppression (negative alpha CTF slope or negative alpha MI) in valid cue sessions. The results from Experiment 2 and Experiment 3 further confirmed the existence of such alpha activity through the negative alpha MI.

Based on the negative CTF slope result in Fig 2, alpha MI during the late period of cue-distractor intervals (see Figs 2, 3, 4, and 6) could also be considered the result of the alpha CTF in the case of lateral cues. As an example, in Experiment 1 (Fig D in S1 Appendix), when the cue pointed left (e.g., $\theta = 288°$), the channel response (alpha power) decreased over the left hemisphere and increased over the right hemisphere; when the cue pointed right ($\theta = 108°$), the channel response (alpha power) increased over the left hemisphere and decreased over the right hemisphere. We calculated such an asymmetric channel response (alpha power) by collapsing across attend-left and attend-right conditions and collapsing across hemispheres (see Materials and methods for more details). The lateralized channel response and observed lateralized alpha power have similar dynamics (compare Figs 2C and 3A) and spatial patterns (Fig D in S1 Appendix). Thus, we inferred that the negative CTF slope might provide a general computational model for the negative alpha MI observed in our study. However, the relationship between CTF and alpha MI needs further study.

A recent study [3] outlined 3 potential computational models for accounting for distractor suppression within the CTF framework. This suggests that distractor-related negative tuning may arise as a consequence of increased tuning towards the opposite distractor direction (model 1), decreased tuning close to the distractor direction (model 2), or a combination of both increased tuning towards the opposite distractor direction and decreased tuning close to the distractor direction (model 3). Through comparison with invalid-cue sessions, our results (Fig 2D) suggest that distractor suppression might result in both tuning towards the opposite distractor direction and away from the cued distractor direction (Fig 2D), which fits well with the interpretations of the above third models. Based on this model, the amplitude and spatial distribution of the alpha band over the scalp has a graded change. Alpha increases were maximal over occipitoparietal electrodes contralateral to the cued distractor location, and maximum alpha decreases were found ipsilaterally to the cued distractor location, so that the to-be-captured resources would be relatively diminished from distractors to support target-related activities. Accordingly, we suggest that during cue-distractor intervals, a template-to-distractor (or spatial priority map) might be architected by the gating role of alpha activity.

By comparing $P_D$ amplitude in Experiment 1, we found that predictable distractors are likely to reduce the amplitude of $P_D$. This result was consistent with previous studies [16,30,42,44], in which predictable distractors reduced distractor-specific processing, as reflected in the decrease of $P_D$. Our results in Experiment 2 further expand this idea and suggest that reduced $P_D$ was not only related to whether the cue was effective or not but also related to whether the predictive validity of the distractor was effective (Fig 4D). Crucially, the correlation across subjects and quartile analysis further showed that reduced $P_D$ amplitude was a function of alpha MI. That is, the more alpha power contralateral to the cued distractor is, the lower the $P_D$ amplitude. Given that a reduced $P_D$ is correlated with minimized distractor interference [56], we argue that the brain can engage in proactive filtering mechanisms that allocate attention resources that are less likely to be deployed to a cued distractor, resulting in less interference by the subsequent distractor.

Note that such transient modulation of alpha power and its link with $P_D$ amplitude does not occur throughout the anticipation period. One possible explanation is that the participants

might strategically have no incentive to persist in suppressing the direction of the task-irrelevant distractor in advance, especially at the cost of task-relevant targets likely occurring in the nearby cued direction. Participants are likely to proactively suppress distractors at a cued location by transient modulation of alpha power. Given that visual and memory systems are reciprocally connected [57–59], alpha power lateralization also reflects spatial inhibition of working memory representations [60], we suspected that a template-to-distractor might not be persistent until the onset of the search display. Alternatively, it was temporarily stored in a visuospatial sketchpad. Once the onset of the visual search was detected, the template-to-distractor can be used to suppress the distractors without feedforward communication of distractor information involving reactive suppression [2]. Our ERP results supported the above hypothesis by showing that smaller $P_D$ followed after a significant negative alpha MI in Experiments 1 and 2. This seems to mean that distractors can be directly suppressed at the low neural level (posterior cortex) in the early stage (approximately 200 ms), resulting in the decrement of distractor-elicited $P_D$ (approximately 200 to 300 ms). Importantly, the significant relationship between transient alpha modulation and $P_D$ amplitude might provide meaningful evidence for the above hypothesis. However, our results showed that significant $P_D$ followed after a significant negative alpha MI in Experiment 3, the decrement of a $P_D$ effect and its link with alpha activity should be interpreted with caution, and further studies are necessary to gain a better understanding of the template-to-distractor that plays a key role in distractor suppression.

In Experiment 1, our ERP results show the distractor-elicited N2pc at approximately 150 ms poststimulus in both valid- and invalid-cue sessions. This finding suggests that participants are first attending to the cued location before suppressing the cued distractor. It is well documented that distractor inhibition may reflect a "search-and-inhibit" mechanism, whereby knowledge of the distractors initially paradoxically increases the attentional bias toward them and then guides attention away [7,15]. Previous studies suggest that once valid information about distractors was provided in advance, people could not immediately escape attraction by the distractor information regardless of task demands [12,15]. An alternative explanation stems from a white bear metaphor that distractor cueing is the visual equivalent of telling someone not to think of a white bear effect [61]. Cueing distractor information is almost paradoxical to instructing someone to prepare to ignore something. They might first try to make an effort to minimize interference from cueing distractor information [11] and then actively inhibit the potential distractor location. In Experiment 2, we only observed the early $P_D$ at approximately 150 ms poststimulus in both high- and low-predictive validity trials but not in null-predictive validity trials. Some studies proposed that early $P_D$ is related to the initial processing of stimuli, which reflects low-level sensory imbalance between the 2 hemispheres [8,62], and the changes of this component could also reflect functional distractor suppression during the early stage [30,51,63]. For null-predictive validity trial, the cue has no information about the visual field where the distractor occurs. Therefore, it is not strange that no early $P_D$ was observed in null-predictive validity. Furthermore, we inferred that no distractor-elicited N2pc, instead of early $P_D$, was observed in Experiment 2, which might also be due to the lack of the specific location where the distractor would appear. Accordingly, the early $P_D$ observed in Experiment 2 might reflect initial and large-scale suppression of the visual field (left visual field or right visual field) where the distractor occurs.

However, the current study only focused on the dynamic characteristics of distractor suppression during anticipation, and the findings are limited in their low-density montages to explain EEG source localization of distractor suppression. Further high-density EEG or MEG is needed to provide a precise source map. In addition, distractors can be suppressed proactively by showing that eye movements are less likely to be deployed to a cued distractor [30]. Eye movement is linked to alpha modulation [30,64] and $P_D$ amplitude [51] during covert

spatial attention. There was no eye-tracking recording in the current study, and only EOG channels were used to measure such movements indirectly. In the future, the combination of eye movement and EEG methods will allow us to investigate both spatial and temporal aspects of distractor inhibition simultaneously.

In summary, cueing the distractor location by spatial cues in different circumstances could influence the distractor spatial proximity effect on behavioral performance and alpha activities. From different guises of alpha activity (CTFs or alpha MI), our results provide insight on the underlying neural mechanisms of proactive suppression, in which alpha power plays an important role in reducing distractor interference. Importantly, a strong link between cue-elicited alpha power and distractor-elicited $P_D$ suggests that alpha power activity may reduce interference following distractor onset. These findings contribute to the growing body of work showing that distractor suppression is flexible and involved in more than one general top-down mechanism [3,34,65].

## Materials and methods

### EEG recording and preprocessing

In all experiments, continuous EEG were recorded using a SynAmps EEG amplifier and Scan 4.5 package (NeuroScan). In Experiment 1, EEG data were recorded from 15 international 10–20 sites, F3, Fz, F4, T3, C3, Cz, C4, T4, P3, Pz, P4, T5, T6, O1, and O2, along with 5 nonstandard sites: OL midway between T5 and O1, OR midway between T6 and O2, PO3 midway between P3 and OL, PO4 midway between P4 and OR, and POz midway between PO3 and PO4. In Experiments 2 and 3, EEG data were recorded using a 32-electrode elastic cap (Greentek) with silver chloride electrodes placed according to the 10–20 system. To detect eye movements and blinks, horizontal electrooculograms (HEOGs) and vertical electrooculograms (VEOGs) were recorded via external electrodes placed at the canthi of both eyes, above and below the right eye, respectively. All electrodes, except those for monitoring eye movements, were referenced to the left mastoid during data collection and then were offline re-referenced to the algebraic average of the left and right mastoids. The EEG were filtered at DC-200 Hz, digitized online at a sampling rate of 1,000 Hz (sampling interval 1 ms), and then offline filtered with a digital bandpass of 0.1 to 40 Hz with cutoff frequency (−6 dB) 0.05 to 40.05 Hz. We kept electrode impedance values below 5 kΩ.

Trials in which the EEG exceeded ±100 μV in any channel and the horizontal EOG exceeded ±50 μV from −200 to 400 ms in the cue- or distractor-locked epochs were automatically excluded in all experiments. Overall, artifacts led to an average rejection rate of 15.4% of trials (range 7.1% to 23.7%) in Experiment 1, 18.0% (range 11.2 to 31.7%) of trials in Experiment 2, and 17.9% of trials (range 8.2% to 29.1%) in Experiment 3. A total of 857 (SD: 49) for each session in Experiment 1, 264 (SD: 28) for each condition in Experiment 2, and 273 (SD: 28) in Experiment 3 were used for further analyses. To assess whether any systematic horizontal EOG activity was present in the remaining data, we computed averaged HEOG waveforms for left- and right-inhibited distractor trials. Residual activity was less than 2 μV in Experiment 1, which means that the residual eye movements were less than ± 0.3˚ [66]. Experiments 2 and 3 consistently demonstrated that the residual activities of HEOG were also less than 2 μV. Thus, the small horizontal eye movements in adults suggest that participants fix their eyes on the center of the screen.

EEG data were preprocessed using the EEGLAB software package in the MATLAB environment [67]. Independent component analysis (ICA, EEGLAB runica function) was performed for continuous data. Component removal was restricted to blink artifacts (less than 2 on average).

## Inverted encoding model analysis

For the IEM analysis, we followed a similar approach to the previous work [36]. We used an IEM to reconstruct location-selective CTFs from the topographic distribution of EEG activity across electrodes to examine the spatially specific alpha-band activity time course. Briefly, this model assumes that the power at each electrode (1 per sample angle) reflects the weighted sum of 10 spatially selective channels [68,69]. We modeled the responses of each electrode using a basis function of 10 half-sinusoids raised to the ninth power for each spatial channel:

$$R = \sin(0.5\theta)^9,$$

such that $\theta$ is the angular location (0˚, 36˚, 72˚, 108˚, 144˚, 180˚, 216˚, 252˚, 288˚, 324˚) and R is the spatial channel response.

EEG data were segmented into 2,000 ms epochs ranging from 500 ms before to 1,500 ms after cue onset for the cue-locked analysis. Data were also segmented and aligned according to target onset from −800 to 800 ms for the distractor-locked analysis. Then, EEG segments were bandpass filtered for the alpha band (8 to 12 Hz) using a function (eegfilt) from the EEGLAB toolbox [67]. The filtered data were transformed to instantaneous power using a function (Hilbert) from MATLAB (The Mathworks, Natick, MA). The IEM was run on each time point in the alpha band power.

We sorted the artifact-free trials into training sets ($B_1$) and test sets ($B_2$) for each subject (for details, see [36]). Let $B_1$ and $B_2$ be the power at each electrode for each trial in the training set and test set, respectively. Data from the training set ($B_1$) were used to estimate channel-to-electrode weights on the hypothetical spatial channels separately for each electrode. The basis functions determined the channel response function ($C_1$) for each spatial channel.

The training data ($B_1$) in electrode space were then mapped onto the matrix of channel outputs ($C_1$) in channel space by the channel-to-electrode weight matrix (W), which was estimated with a general linear model of the form:

$$B_1 = WC_1$$

The estimated channel-to-electrode weight matrix can be derived via least-squares estimation as follows:

$$\widehat{W} = B_1 C_1^T (C_1 C_1^T)^{-1}$$

In the test stage, channel responses ($C_2$) were estimated based on the observed test data ($B_2$) with the weight matrix W:

$$C_2 = (\widehat{W}^T \widehat{W})^{-1} \widehat{W}^T B_2.$$

Finally, the 10 estimated response functions ($C_2$) were aligned to a common center. The center channel was the channel tuned for the location of the specific stimulus (that is, $\theta$˚) and then averaged to obtain the CTF. The CTF slope was used as a metric to compare attention deployment towards the distractor.

## Alpha modulation analysis

The segmented EEG data were decomposed using Morlet wavelet-based analysis from 8 to 12 Hz in 1 Hz steps implemented in the related package Brainstorm [70] in the MATLAB environment. Differences in time-frequency power could be explained by event-related differences in ongoing oscillatory power (that is, "induced" activity) or ERP exhibiting strong power in (low-) frequency components ("evoked" activity; [71,72]). To evaluate this issue, we separated

nonphase-locked from phase-locked power. We subtracted the trial-average activity in the time domain from the EEG activity of every single trial, thus effectively removing any phase-locked component, so we could be certain that the resulting oscillatory activity was nonphase-locked, which was not contaminated by ERPs. This method was also widely used in recent EEG studies [73,74].

To estimate the effects of cue-elicited attention modulation, we calculated the alpha MI from cue-locked data for 3 pairs of parietal and occipital electrodes (left ROI: P3, P7, O1; right ROI: P4, P8, O2). The MI was computed using the following formula:

$$\text{Alpha MI} = \left( \sum_\theta^n \frac{\alpha_{Left\ ROI}^\theta - \alpha_{Left\ ROI}^{\theta-180}}{\frac{1}{2}\left(\alpha_{Left\ ROI}^\theta + \alpha_{Left\ ROI}^{\theta-180}\right)} - \sum_\theta^n \frac{\alpha_{Right\ ROI}^\theta - \alpha_{Right\ ROI}^{\theta-180}}{\frac{1}{2}\left(\alpha_{Right\ ROI}^\theta + \alpha_{Right\ ROI}^{\theta-180}\right)} \right),$$

where θ indicates the angle of cue pointing (θ = 288˚ or 252˚ in Experiments 1; θ = 270˚ in Experiments 2 and 3); α indicates alpha band power within the left ROI or right ROI; and n is the number of θ in the modulation analysis ($n$ = 2 in Experiment 1; $n$ = 1 in Experiments 2 and 3).

Note that the above method allowed us to avoid possible bias in the analysis due to the hemisphere asymmetry [31]. The amplitude of MI denotes the deviation of spatial alpha power in the hemisphere contralateral to the cued distractor with respect to the hemisphere ipsilateral to the distractor. Further, the polarity of alpha MI denotes the direction of spatial alpha modulation: Positive values indicate alpha power relatively decreases contralateral to distractor; negative values indicate alpha power relatively increases contralateral to the distractor.

## Decoding analysis

We adopted the same procedure as reported in a previous study [3], except with 22 EEG channels as features and the spatial location of distractors or targets as classes. In brief, we used multivariate pattern analysis (MVPA) in combination with linear discriminant analysis to assess whether the spatial distribution of EEG data could be used to decode the distractor or target location in Experiment 1. The decoding algorithm employed was linear discriminant analysis [75], which is consistent with one previous study [50]. The performance of decoding based on EEG data is the 10-fold cross-validation area under the ROC curve (AUC) of the corresponding model.

## ERP analysis

The EEGLAB toolbox and ERPLAB toolbox [76] were used to process and analyze ERP. The combination of a lateral distractor and a midline target (see Fig 3) enables the isolation of EEG activity in response to the distractor [23]. Thus, we analyzed the ERP elicited by the subsequent visual search display with a lateral distractor and midline target to isolate distractor-specific $P_D$ components. ERP was computed by subtracting the waveforms measured from electrodes (P7 or P8) on the ipsilateral hemisphere to the distractor from symmetrical electrodes on the contralateral hemisphere. Then, ERP was corrected using a −200 to 0 ms window preceding stimulus onset. All latencies were identified using standard cluster-based nonparametric tests with a cluster-defining threshold of $p < 0.05$ [4,77]. Finally, the amplitude of $P_D$ was achieved in the ERPLAB measurement tool as the mean value of a 20-ms window centered at the most positive peak in the averaged difference waveform.

## Correlation and quartile analysis

We performed a similar time-frequency correlation method reported in Zhao's research [78]. We extracted alpha MI values based on a 60-ms sliding time window (steps of 5 ms) across a

time range of −200 to 1,200 ms for each subject and then correlated them with distractor-evoked $P_D$. Each pixel of the time-frequency correlation map consisted of Pearson's r value between alpha MI at each time interval and each frequency and subsequent $P_D$ amplitude. Then, the significant spectrogram related to $P_D$ amplitude ($p < 0.050$) was corrected for false discovery rates (FDRs) within a prior defined frequency range of 8 to 12 Hz across the entire time. The left significant spectrogram ($p_{corrected} < 0.050$) was defined as the TFC ROI.

We also adopted a similar quartile analysis within subjects as reported in van Dijk's research [52]. The average single-trial $P_D$ was estimated at the within-subjects level to confirm the relationship between the alpha MI and subsequent $P_D$ amplitude. The trials were sorted according to alpha MI and split into quartiles for the right-attend and left-attend session. The separate $P_D$ waveforms for each session were calculated for each quartile and normalized to the individual mean value over all quartiles. The final $P_D$ for each quartile was computed by averaging the $P_D$ from the right- and left-attend sessions.

### Participants

A total of 110 paid volunteers participated in the 3 EEG experiments (Experiment 1: 32, Experiment 2: 28, Experiment 3: 50), 12 of whom were excluded from statistical analysis due to excessive EEG artifacts with rejected trials >30% (2 participants in Experiment 1; 2 participants in Experiment 2; 8 participants in Experiment 3). Data from the remaining 30 participants in Experiment 1 (12 male, 18 female, 22.6 years mean age), 26 participants in Experiment 2 (9 male, 17 female, 22.7 years mean age), and 42 participants in Experiment 3 (12 male, 30 female, 23.2 years mean age) were used. Additional 20 participants (8 male, 12 female, 21.5 years mean age) paid volunteers participated in the behavioral control experiment. All participants had a normal or corrected-to-normal vision and were right-handed. They were neurologically unimpaired and gave informed written consent before the experiment. All experiments were approved by State Key Laboratory of Cognitive Neuroscience and Learning Institutional Review Board (dossier number IRB_B_0016_2015002). All participants gave written informed consent. The study was in full compliance with the ethical practice of Beijing Normal University.

### Task, stimuli, and procedure

Previous studies [3,7] have arranged target and distractor locations by dividing 2D space into 4 or 6 parts. It is likely that within these tasks, spatial distractor cues might indirectly provide potential spatial information about a target, e.g., when the distractor was occurring on the left, the target was presented on the right more often, and vice versa. Considering that participants pick up such statistical regularities and use them to guide their target selection [7,79], increased alpha power contralateral to the distractor might be mixed by potential target-related activity (decreased alpha power contralateral to more often the target). Thus, ensuring that participants do not have target-related activity is essential to study distractor suppression, which is also in compliance with the relevant principles (see rule 2 in [80]). In this sense, we minimized target-dependent activity by increasing the number of possible directions ($N$ = 10) and decreasing the probability of the target occurring on the lateral side (Experiments 2 and 3).

In this study, 3 experiments were conducted to investigate the influences of the spatial cues of the distractor on the subsequent visual search. In each experiment, a 200-ms cue informed the participants of the location (Experiment 1) or scope (Experiments 2 and 3) in which the upcoming distractor would occur in the search display. The cue-distractor interval was 1,200 to 1,600 ms. Each search display consisted of 10 unfilled circles presented for 200 ms (13.5 cd/m$^2$ mean optical luminance, and 3.4° × 3.4°, 0.3° thick outline) from the imaginary ring with a

9.2˚ radius. A yellow target circle and a red distractor circle were simultaneously presented among the 8 green circles. A schematic of the trial design is illustrated in Fig 1A.

Salience was defined in terms of the local contrast between green circles and each color circle (see Fig E in S1 Appendix): The distance in chromaticity space between the red distractor and green circles was greater than the distance between the yellow target circle and green circles. A red distractor with more salience captured attention more easily than a yellow target, creating more incentive to ignore distracting sensory information. Before each session, participants were informed of the cue validity (valid or invalid) and its corresponding spatial probability of the target and distractor (Fig 1B). Participants completed some practice trials to ensure that they understood the task requirements and learned how to make good use of cues. Participants were instructed to utilize the cue to ignore a more salient distractor and determine whether the line segment inside the target (yellow circle) was vertical or horizontal by pressing 1 of 2 buttons with their right hand as quickly as possible.

## Experiment 1

In Experiment 1, a red circular sector with an angle of 36˚ was embedded in a full green circle at the center of the display (see Fig 1A), which randomly and equally pointed to 1 of 10 possible directions (0˚, 36˚, 72˚, 108˚, 144˚, 180˚, 216˚, 252˚, 288˚, or 324˚) with reference to the upper y-axis (0˚). As shown in Fig 1B, this graphic cue was typically informative for the valid-cue session (100% probability on a cued location) or uninformative for the invalid-cue session (10% probability on a cued location) of the location at which the subsequent red distractor circle emerged. In both valid- and invalid-cue sessions, the location of the subsequent target was independent of which distractor location and randomized with equal probability (10% probability on each location), so that subjects could not infer anything about the yellow target circle from the cue. The sequence of the 2 sessions was counterbalanced between subjects. Each session consisted of ten 100-trial blocks and lasted approximately 60 minutes. Participants came to the lab twice, separated by 1 week.

In the behavioral control experiment, the apparatus, stimuli, and procedure were the same as in Experiment 1, except for the following: (1) We replaced the red distractor with a blue distractor in half of the blocks. It is noted that the blue distractor has the same salience as the red distractor, as the distance in chromaticity space between the red distractor and green circles was equal to the distance between the blue distractor (RGB: 45, 0, 245) circle and green circles. (2) We removed salient distractors in 20% of trials (distractor-absent trials) in both valid- and invalid- cue sessions. That is, in the remaining 80% of distractor-present trials, the cue display was fully predictive or not predictive of the specific location of the red or blue distractor circle in 2 separate sessions.

## Experiment 2

There were 3 kinds of graphic cues in Experiment 2. The circular sector was equally likely to point left (90˚) or right (270˚), and the variable area of the circular sector was related to the predictive validity of distractor occurrence. As shown in spatial probability in Fig 4A, (1) in the high-predictive validity trials, the red sector with a polar angle from 216˚ to 324˚ (or from 36˚ to 144˚) was fully predictive with 100% validity for the left (or right) side where the red circle distractor would appear, that is, the distractor would appear randomly on one of the cued lateral locations with 25% probability; (2) in the low-predictive validity trials, a red semicircle predicted that the red circle distractor would appear randomly on one of the cued locations (with 16.7% probability on one lateral location or one midline location); (3) in the null-predictive validity trials, none of the red sectors embedded in the green circle was uninformative of

the upcoming distractor (10% probability on each location). To isolate the brain activity related to distractor anticipation, we pseudorandomized the location of a yellow target circle by specifying a uniform spatial probability of 4.25% on each lateral location and 33% on each midline location (see Fig 4A; right panel). The experiment contained 10 blocks (that is, 100 trials per block) per participant. The 3 types of trials were randomized within each block. Experiment 2 lasted approximately 60 minutes.

### Experiment 3

In Experiment 3, we used the constant arrow instead of the variable circular sector as a symbolic spatial cue (see Fig 6A, left panel). The red arrow was fully predictive of the side (with 100% validity) on which the following red distractor circle would subsequently appear, that is, the distractor would appear randomly on one of the cued lateral locations with 25% probability (Fig 6A, middle panel). The opposite green arrow had no predictive value for the yellow target circle and red distractor circle. Target had the same spatial probability as that of Experiment 2 (Fig 6A, right panel). In fact, the cue in Experiment 3 was the same as the high-predictive validity trials in Experiment 2 except for the symbolic form of a spatial cue. We called it the arrow high-predictive validity cue.

## Supporting information

**S1 Appendix. Supporting information.** Section A. Mean performance for Experiments 1 and 2. Section B. Decoding for distractor and target. Section C. The analysis pipeline for CTF analysis and alpha MI. Section D. Colour space in the present study.
(PDF)

**S1 Data. Raw data file for Fig 1C–1H.**
(XLSX)

**S2 Data. Raw data file for Fig 2C–2E.**
(XLSX)

**S3 Data. Raw data file for Fig 3A–3C.**
(XLSX)

**S4 Data. Raw data file for Fig 4B–4E.**
(XLSX)

**S5 Data. Raw data file for Fig 5A–5D.**
(XLSX)

**S6 Data. Raw data file for Fig 6B–6E.**
(XLSX)

**S7 Data. Raw data file for Fig A in S1 Appendix.**
(XLSX)

**S8 Data. Raw data file for Fig B in S1 Appendix.**
(XLSX)

**S9 Data. Raw data file for Fig C in S1 Appendix.**
(XLSX)

**S10 Data. Raw data file for Fig D in S1 Appendix (heatmaps).**
(XLSX)

**S11 Data. Raw data file for Fig E in S1 Appendix.**
(XLSX)

## Author Contributions

**Conceptualization:** Chenguang Zhao, Ole Jensen, Yan Song.

**Data curation:** Yuanjun Kong, Lujiao Kong.

**Formal analysis:** Chenguang Zhao, Yuanjun Kong.

**Funding acquisition:** Chenguang Zhao, Xiaoli Li, Yan Song.

**Investigation:** Chenguang Zhao, Yuanjun Kong.

**Methodology:** Chenguang Zhao, Yuanjun Kong, Dongwei Li.

**Software:** Chenguang Zhao.

**Supervision:** Jing Huang, Xiaoli Li, Ole Jensen, Yan Song.

**Validation:** Dongwei Li.

**Visualization:** Chenguang Zhao, Lujiao Kong.

**Writing – original draft:** Chenguang Zhao, Yuanjun Kong.

**Writing – review & editing:** Yuanjun Kong, Dongwei Li, Jing Huang, Lujiao Kong, Xiaoli Li, Ole Jensen, Yan Song.

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
