## [Editor Report · Decision Letter 0]

16 Nov 2022

Dear Dr Song, 

Thank you for submitting your revised manuscript entitled "Neural mechanisms underlying distractor suppression guided by visual-spatial cues" for consideration as a Research Article by PLOS Biology.

Before we can re-evaluate the revisions made to your prior submission (PBIOLOGY-D-22-01312), we need you to complete your submission by providing the metadata that is required for full assessment. To this end, please login to Editorial Manager where you will find the paper in the 'Submissions Needing Revisions' folder on your homepage. Please click 'Revise Submission' from the Action Links and complete all additional questions in the submission questionnaire.

Once your full submission is complete, your paper will undergo a series of checks. After your manuscript has passed the checks it will be sent back to me. To provide the metadata for your submission, please Login to Editorial Manager (https://www.editorialmanager.com/pbiology) within two working days, i.e. by Nov 18 2022 11:59PM.

Kind regards,

Kris

Kris Dickson, Ph.D., (she/her)

Neurosciences Senior Editor/Section Manager

PLOS Biology

kdickson@plos.org

---

## [Decision Letter · Decision Letter 1]

12 Jan 2023

Dear Dr Song,

Thank you for your patience while we considered your revised manuscript "Neural mechanisms underlying distractor suppression guided by visual-spatial cues" for publication as a Research Article at PLOS Biology. This revised version of your manuscript has been evaluated by the PLOS Biology editors, the Academic Editor and one of the original reviewers (Reviewer 3 on PBIOLOGY-D-22-01312).

Based on this reviewer's feedback and our Academic Editor's assessment of your revision, we are likely to accept this manuscript for publication, provided you satisfactorily address the remaining points raised by this reviewer.

***When revising, we'd ask that you also consider a minor title change to ensure the work is as broadly accessible to our audience as possible. We'd suggest:

"Suppression of distracting inputs by visual-spatial cues is driven by anticipatory alpha activity"

***Please also make sure to address the data and other policy-related requests listed below my signature. Note that failure to fully address these points will delay moving forward with this work.

We expect to receive your revised manuscript within two weeks. 

*Published Peer Review History*

*Press*

Sincerely,

Kris

Kris Dickson, Ph.D., (she/her)

Neurosciences Senior Editor/Section Manager,

kdickson@plos.org,

PLOS Biology

****BLURB:

Please also provide a blurb which, if the paper is accepted, will be included in our weekly and monthly Electronic Table of Contents (eTOCs), sent out to readers of PLOS Biology. This blurb may also be used to promote your article on social media. The blurb should be about 30-40 words long and is subject to editorial changes. It should, without exaggeration, entice people to read your manuscript, should not be redundant with the title and should not contain acronyms or abbreviations. For examples, view our author guidelines: https://journals.plos.org/plosbiology/s/revising-your-manuscript#loc-blurb

DATA POLICY:

Thank you for supplying the underlying data for this work in compliance with the PLOS Data Policy, which requires that all data be made available without restriction: http://journals.plos.org/plosbiology/s/data-availability. For more information, please also see this editorial: http://dx.doi.org/10.1371/journal.pbio.1001797. Note that we do not require all raw data. Rather, we ask that all individual quantitative observations that underlie the data summarized in the figures and results of your paper be made available.

1) Please ensure that all data files are uinvariably referred to (in the manuscript, figure legends, and the Description field when uploading your files) using the following format verbatim: S1 Data, S2 Data, etc. Multiple panels of a single or even several figures can be included as multiple sheets in one excel file that is saved using exactly the following convention: S1_Data.xlsx (using an underscore). This will make it easier for our readership to navigate this data and reproduce your results.

2) Please ensure that you have provided the individual numerical values that underlie the summary data displayed in the following figure panels as they are essential for readers to assess your analysis and to reproduce it:

Fig1C-H; Fig2C-E; Fig3a-C; Fig4B-E; Fig5A-D; Fig6B-E;

Supplemental FigS1; FigS2A-F; FigS3; FigS4B,C heatmaps; FigS5

3) Please also ensure that figure legends in your manuscript include information on where the underlying data can be found (e.g. The data in Fig 1C can be found at XXX labeled YYY), and ensure your supplemental data file/s has a legend.

DATA NOT SHOWN?

- Please note that per journal policy, we do not allow the mention of "data not shown", "personal communication", "manuscript in preparation" or other references to data that is not publicly available or contained within this manuscript. Please check your work carefully and either remove mention of any such data or provide figures presenting the results and the data underlying the figure(s).

Reviewer remarks:

Reviewer's Responses to Questions

Do you want your identity to be public for this peer review?

Reviewer #1: Yes: Clayton Hickey

Reviewer #1: The authors have largely addressed the concerns I raised in prior review. I identify a few residual issues below but these are all minor in nature. 

p.21 '…the cue is more effective when it predicts distractors further away from the target.' This is at odds with what we found in van Zoest et al. 2021. There, the cue was effective at reducing misdirection of the eyes to the distractor only when the target and distractor were in close spatial proximity (fig 2). There might be the opportunity to mention this disparity, note that the difference might result from the large difference in raw latency of eye movements versus manual response. 

p.21 - '…by demonstrating that the spatial proximity of the distractor [effect] was influenced not only by statistical learning but also by the spatial cues.' This was also demonstrated in van Zoest et al. 2021, though the effect there was in the opposite direction. 

P26 - 'Given that the early Pd is related to… it is assumed to reflect low-level sensory imbalance…' That's not entirely accurate. There can be sensory imbalance in the latency of the early Pd, but the component can clearly have functional significance eg. Sawaki & Luck, 2010 APP; Weaver, van Zoest, & Hickey, 2017 Neuroimage. This line cites van Zoest et al., 2021, where the early Pd is shown to have functional significance, so the statement conflicts with the citation. 

Minor: 

p.5 23 - 'We hypothesized that cueing the distractor location would influence on the spatial proximity of the distractor.' The effect is not on the spatial proximity of the distractor, but rather on how the spatial proximity of the distractor impacts another measure like behaviour or alpha. Strictly read, this line tells the reader that the cue changes the distance between target and distractor, which is not the intent. Also bottom p 7, top p.9, p.11 10, p21 17, elsewhere in MS. 

P 4 9 - '…several recent studies have focused on the investigation of alpha power in terms of its hemispheric lateralization and spatial selectivity.' 

P 5 9 - '…albeit the trial-wise magnitude of the pre-target alpha power has been linked to lateral indices of attentional selection in the ERP [30].'

- There's room for more description of the cluster statistics. In particular, what was the cluster- defining statistical threshold? 

p.13 9 - Citation here to Cunningham & Egeth is wrong, there is no reference to N2pc in that paper.

---

## [Editor Report · Decision Letter 2]

27 Jan 2023

Dear Dr Song,

Thank you for the submission of your revised Research Article "Suppression of distracting inputs by visual-spatial cues is driven by anticipatory alpha activity" for publication in PLOS Biology. On behalf of my colleagues and the Academic Editor, Simon Hanslmayr, I am pleased to say that we can in principle accept your manuscript for publication, provided you address any remaining formatting and reporting issues. These will be detailed in an email you should receive within 2-3 business days from our colleagues in the journal operations team; no action is required from you until then. Please note that we will not be able to formally accept your manuscript and schedule it for publication until you have completed any requested changes.

Please also provide a blurb which, if the paper is accepted, will be included in our weekly and monthly Electronic Table of Contents (eTOCs), sent out to readers of PLOS Biology. This blurb may also be used to promote your article on social media. The blurb should be about 30-40 words long and is subject to editorial changes. It should, without exaggeration, entice people to read your manuscript, should not be redundant with the title and should not contain acronyms or abbreviations. For examples, view our author guidelines: https://journals.plos.org/plosbiology/s/revising-your-manuscript#loc-blurb

PRESS

We frequently collaborate with press offices. If your institution or institutions have a press office, please notify them about your upcoming paper at this point, to enable them to help maximize its impact. If the press office is planning to promote your findings, we would be grateful if they could coordinate with biologypress@plos.org. If you have previously opted in to the early version process, we ask that you notify us immediately of any press plans so that we may opt out on your behalf.

Sincerely, 

Kris

Kris Dickson, Ph.D., (she/her)

Neurosciences Senior Editor/Section Manager

PLOS Biology

kdickson@plos.org